# Stochastic fluctuations of bosonic dark matter

Gary P. Centers [1,2], John W. Blanchard [2], Jan Conrad [3], Nataniel L. Figueroa [1,2], Antoine Garcon[1,2], Alexander V. Gramolin [4], Derek F. Jackson Kimball [5], Matthew Lawson [2,3], Bart Pelssers[3], Joseph A. Smiga [1,2], Alexander O. Sushkov [4], Arne Wickenbrock [1,2], Dmitry Budker [1,2,6✉] & Andrei Derevianko [7]

Numerous theories extending beyond the standard model of particle physics predict the existence of bosons that could constitute dark matter. In the standard halo model of galactic dark matter, the velocity distribution of the bosonic dark matter field defines a characteristic coherence time $\tau_c$. Until recently, laboratory experiments searching for bosonic dark matter fields have been in the regime where the measurement time $T$ significantly exceeds $\tau_c$, so null results have been interpreted by assuming a bosonic field amplitude $\Phi_0$ fixed by the average local dark matter density. Here we show that experiments operating in the $T \ll \tau_c$ regime do not sample the full distribution of bosonic dark matter field amplitudes and therefore it is incorrect to assume a fixed value of $\Phi_0$ when inferring constraints. Instead, in order to interpret laboratory measurements (even in the event of a discovery), it is necessary to account for the stochastic nature of such a virialized ultralight field. The constraints inferred from several previous null experiments searching for ultralight bosonic dark matter were overestimated by factors ranging from 3 to 10 depending on experimental details, model assumptions, and choice of inference framework.

[1] Johannes Gutenberg-Universität, Mainz 55128, Germany. [2] Helmholtz Institute, Mainz 55099, Germany. [3] Department of Physics, Stockholm University, AlbaNova, 10691 Stockholm, Sweden. [4] Department of Physics, Boston University, Boston, MA 02215, USA. [5] Department of Physics, California State University East Bay, Hayward, CA 94542-3084, USA. [6] Department of Physics, University of California, Berkeley, CA 94720-7300, USA. [7] Department of Physics, University of Nevada, Reno, NV 89557, USA. ✉email: budker@uni-mainz.de

It has been nearly ninety years since strong evidence of the missing mass we label today as dark matter (DM) was revealed[1], and its composition remains one of the most important unanswered questions in physics. There have been many DM candidates proposed and a broad class of them, including scalar (dilatons and moduli[2–5]) and pseudoscalar particles (axions and axion-like particles[6–11]), can be treated as an ensemble of identical bosons, with statistical properties of the corresponding fields described by the standard halo model (SHM)[12,13]. In this work, our model of the resulting bosonic field assumes that the local DM is virialized and neglects non-virialized streams of DM[14], Bose–Einstein condensate formation[15–18], and possible small-scale structure such as miniclusters and axion stars[19–21]. To date, it is typical to ignore such DM structure when calculating experimental constraints, and within this isotropic SHM DM model, we demonstrate the general weakening of inferred constraints due to the statistical properties of the virialized ultralight field (VULF)[21–24]. We note that some astrophysical and cosmological simulations can and do resolve these stochastic properties[25,26], however in this paper we discuss their impact on inferences drawn from direct detection experiments.

During the formation of the Milky Way the DM constituents relax into the gravitational potential and obtain, in the galactic reference frame, a velocity distribution with a characteristic dispersion (virial) velocity $v_{\mathrm{vir}} \approx 10^{-3}c$ and a cut-off determined by the galactic escape velocity. Following Refs. [27,28] we refer to such virialized ultralight fields, $\phi(t, \boldsymbol{r})$, as VULFs, emphasizing their SHM-governed stochastic nature. Neglecting motion of the DM, the field oscillates at the Compton frequency $f_c = m_\phi c^2 h^{-1}$. However, there is broadening due to the SHM velocity distribution according to the dispersion relation for massive non-relativistic bosons: $f_\phi = f_c + m_\phi v^2 (2h)^{-1}$. The field modes of different frequency and random phase interfere with one another resulting in a net field exhibiting stochastic behavior. The dephasing of the net field can be characterized by the coherence time $\tau_c \equiv \left(f_c v_{\mathrm{vir}}^2/c^2\right)^{-1}$ [29]. We note that there is some ambiguity in the definition of the coherence time, up to a factor of $2\pi$, and adopt that which was used in the majority of the literature. See the discussion in Supplementary Note 4.

While the stochastic properties of similar fields have been studied before, for example in the contexts of statistical radiophysics, the cosmic microwave background, and stochastic gravitational fields[30], the statistical properties of VULFs have only been explored recently. The 2-point correlation function, $\langle \phi(t, \mathbf{r})\phi(t', \mathbf{r}')\rangle$, and corresponding frequency-space DM "lineshape" (power spectral density, PSD) were derived in Ref. [28], and rederived in the axion context by the authors of ref. [31]. While refs. [28,31] explicitly discuss data-analysis implications in the regime of the total observation time $T$ being much larger than the coherence time, $T \gg \tau_c$, detailed investigation of the regime $T \ll \tau_c$, until now, has been lacking (although we note that ref. [31] includes a brief discussion of the change in sensitivity due to coherent averaging for this regime in their Appendix E). Note that a preprint of this paper has been available online since 2019, and multiple experimental groups have already used it to correct their exclusion limits for stochastic fluctuations or noted the effect[32–41].

We focus on this regime, $T \ll \tau_c$, characteristic of experiments searching for ultralight (pseudo)scalars with masses $\lesssim 10^{-13}$ eV[42–48] that have field coherence times $\gtrsim 1$ day. This mass range is of significant interest as the lower limit on the mass of an ultralight particle extends to $10^{-22}$ eV and can be further extended if it does not make up all of the DM[49]. Additionally, there has been recent theoretical motivation for "fuzzy dark matter" in the $10^{-22}$–$10^{-21}$ eV range[23,49–53], and the so-called string "axiverse" extends to $10^{-33}$ eV[54]. Similar arguments also apply to dilatons and moduli[55].

Here, we show that for experiments operating in the $T \ll \tau_c$ regime it is incorrect to assume a fixed value of $\Phi_0$ when inferring constraints on the coupling strength of bosonic DM to standard-model particles. The constraints inferred from several previous null experiments searching for ultralight bosonic DM were overestimated by factors ranging from 3 to 10 depending on experimental details, model assumptions, and choice of inference framework.

## Results

**Model of bosonic dark matter and amplitude distribution.** Figure 1 shows a simulated VULF field, illustrating the amplitude modulation present over several coherence times. At short time scales ($\ll \tau_c$), the field coherently oscillates at the Compton frequency, see the inset of Fig. 1, where the amplitude $\Phi_0$ is fixed at a single value sampled from its distribution. An unlucky experimentalist could even have near-zero field amplitudes during the course of their measurement.

On these short time scales, the DM signal $s(t)$ exhibits a harmonic signature,

$$s(t) = \gamma \xi \phi(t) \approx \gamma \xi \Phi_0 \cos(2\pi f_\phi t + \theta) , \qquad (1)$$

where $\gamma$ is the coupling strength to a standard-model field and $\theta$ is an unknown phase. Details of the particular experiment are accounted for by the factor $\xi$. In this regime, the amplitude $\Phi_0$ is unknown and yields a time-averaged energy density $\langle \phi(t)^2 \rangle_{T \ll \tau_c} = \Phi_0^2/2$. However, for times much longer than $\tau_c$ the energy density approaches the ensemble average determined by $\langle \Phi_0^2 \rangle = \Phi_{\mathrm{DM}}^2$. This field oscillation amplitude is estimated by assuming that the average energy density in the bosonic field is equal to the local DM energy density $\rho_{\mathrm{DM}} \approx 0.4\,\mathrm{GeV/cm^3}$, and thus $\Phi_{\mathrm{DM}} = \hbar (m_\phi c)^{-1}\sqrt{2\rho_{\mathrm{DM}}}$.

The oscillation amplitude sampled at a particular time for a duration $\ll \tau_c$ is not simply $\Phi_{\mathrm{DM}}$, but rather a random variable whose sampling probability is described by a distribution characterizing the stochastic nature of the VULF. Until recently, most experimental searches have been in the $m_\phi \gg 10^{-13}$ eV regime with short coherence times $\tau_c \ll 1$ day[56–70]. However, for smaller boson masses it becomes impractical to sample over many coherence times: for example, $\tau_c \gtrsim 1$ year for $m_\phi \lesssim 10^{-16}$ eV. Assuming the value $\Phi_0 = \Phi_{\mathrm{DM}}$ neglects the stochastic nature of the bosonic dark matter field[42–48].

The net field $\phi(t)$ is a sum of different field modes with random phases. The oscillation amplitude, $\Phi_0$, results from the inter-

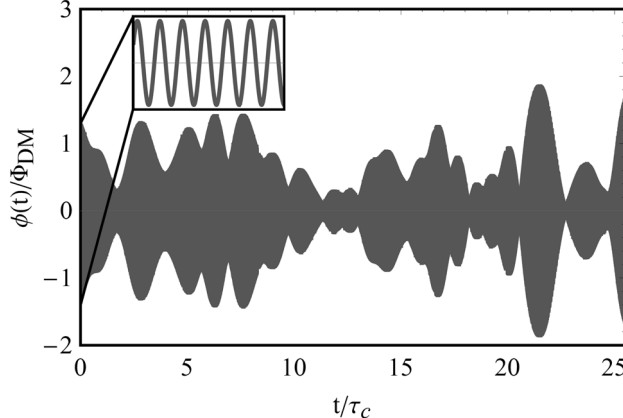

**Fig. 1 Simulated VULF based on the approach in ref. [28] with field value $\phi(t)$ and time normalized by $\Phi_{\mathrm{DM}}$ and coherence time $\tau_c$, respectively.** The inset plot displays the high-resolution coherent oscillation starting at $t = 0$.

ference of these randomly phased oscillating fields. This can be visualized as arising from a random walk in the complex plane, described by a Rayleigh distribution[31]

$$p(\Phi_0) = \frac{2\Phi_0}{\Phi_{DM}^2} \exp\left(-\frac{\Phi_0^2}{\Phi_{DM}^2}\right), \qquad (2)$$

analogous to that of chaotic (thermal) light[71]. This distribution implies that ≈63% of all amplitude realizations will be below the r.m.s. value $\Phi_{DM}$. Equation (2)[31] is typically represented in its exponential form[72] (Supplementary Note 3), and is well sampled in the $T \gg \tau_c$ regime. However, this stochastic behavior should not be ignored in the opposite limit. Simulations of galactic $p(\Phi_0)$ distributions for fuzzy dark matter show slightly heavier tails than the random phase model[35], Eq. (2), but these differences have a negligible effect on the results of this paper as we discuss in Supplementary Note 5.

**Establishing constraints on coupling strength**. We refer to the conventional approach assuming $\Phi_0 = \Phi_{DM}$ as deterministic and approaches that account for the VULF amplitude fluctuations as stochastic. To compare these two approaches we choose a Bayesian framework and calculate the numerical factor affecting coupling constraints, allowing us to illustrate the effect on exclusion plots of previous deterministic constraints[42–48]. It is important to emphasize that different frameworks to interpret experimental data than presented here can change the magnitude of this numerical factor[73–76], see Supplementary Note 1 for a detailed discussion. In any case, accounting for this stochastic nature will generically relax existing constraints as we show below.

We follow the Bayesian framework[77] (see application to VULFs in ref. [28]) to determine constraints on the coupling-strength parameter $\gamma$. Bayesian inference requires prior information on the parameter of interest to derive its respective posterior probability distribution, in contrast to purely likelihood-based inference methods. The central quantity of interest in our case is the posterior distribution for possible values of the coupling constant $\gamma$, derived from Bayes theorem,

$$p(\gamma|D, f_\phi, \xi) = \mathcal{C} \int p(\gamma, \Phi_0) \mathcal{L}(D|\gamma, \Phi_0, f_\phi, \xi) d\Phi_0. \qquad (3)$$

The left-hand side of the equation is the posterior distribution for $\gamma$, where $D$ represents the data, and the Compton frequency $f_\phi$ is a model parameter. $\mathcal{C}$ is the normalization constant, and the likelihood $\mathcal{L}(\cdots)$ is the probability of obtaining the data $D$ given that the model and prior information, such as those provided by the SHM, are true. The integral on the right-hand side accounts for (marginalizes over) the unknown VULF amplitude $\Phi_0$, which we assume follows the Rayleigh distribution described by Eq. (2). For the choice of prior $p(\gamma, \Phi_0)$ we use what is known as an objective prior[78]: the Berger–Bernardo reference prior[79]. Note that this approach is equivalent to starting with the marginal likelihood $\int d\Phi_0 p(\Phi_0) \mathcal{L}(\cdots)$ and using Jefferey's prior to calculate the posterior[80]. See details in Supplementary Note 1.

Results from Bayesian inference are sensitive to the choice of prior[79], and we find better agreement with frequentist-based approaches when using an objective prior rather than a uniform prior $p(\gamma) = 1$ (as shown in Supplementary Note 1). Additionally, the uniform prior yields constraints that are noninvariant under a change of variable.

It is important to note that experiments searching for couplings of VULFs to fermion spins (axion "wind" searches) are sensitive to the projection of the field gradient onto the sensitive axis of the experiment. Due to this directional sensitivity, the derived coupling strength strongly depends on specific experimental

conditions. However, under some reasonable assumptions discussed in Supplementary Note 1, the correction factor is similar in size to the scalar case considered here. Axion-wind experiments can also utilize the daily modulation of this projection, due to rotation of the Earth, to search for signals with an oscillation period much longer than the measurement time $T \ll 1/f_\phi$. The unknown initial phase $\theta$ of the VULF sets the amplitude of this daily oscillation and also needs to be marginalized over. We discuss these topics in Supplementary Note 1, relevant for the experiments[42–45], and focus solely on stochastic variations of the scalar field amplitude, $\Phi_0$, here.

Using the posterior distribution, $p(\gamma|D, f_\phi, \xi)$, one can set constraints on the coupling strength $\gamma$. Such a constraint at the commonly employed 95% confidence level (CL), $\gamma_{95\%}$, is given by

$$\int_0^{\gamma_{95\%}} p(\gamma|D, f_\phi, \xi) d\gamma = 0.95. \qquad (4)$$

The posteriors in both the deterministic and stochastic treatments are derived in Supplementary Note 1. In short, the two posteriors differ due to the marginalization over $\Phi_0$ for the stochastic case, see the integral of Eq. (3). Assuming white noise of variance $\sigma^2$ and that the data are in terms of excess amplitude $A$ (observed Fourier amplitude divided by expected noise, an analog to the excess power statistic) we can derive the posterior for excess signal amplitude $A_s$. The posteriors are

$$p_{det}(A_s|A) \propto p(A_s) 2A \exp\left(-A^2 - A_s^2\right) I_0\left(2AA_s\right), \qquad (5)$$

$$p_{stoch}(A_s|A) \propto p(A_s) \frac{2A}{(1+A_s^2)} \exp\left(-\frac{A^2}{1+A_s^2}\right). \qquad (6)$$

Here $A_s \equiv \gamma \times \xi \Phi_{DM} \sqrt{N}/(2\sigma)$, $I_0(x)$ is the modified Bessel function of the first kind, and $p(A_s)$ is effectively the prior on $\gamma$. In Fig. 2, we plot the normalized posteriors assuming $A$ at the 95% detection threshold $A^{dt} = \sqrt{-\ln(1 - 0.95)}$ and using Berger–Bernardo reference priors for $p(A_s)$; we compare other choices of prior in Supplementary Note 1. The derivation relies on the discrete Fourier transform for a uniform sampling grid of $N$ points and the assumptions of the uniform grid and white noise can be relaxed[28].

Examination of Eqs. (5), (6) and Fig. 2 reveals that the fat-tailed stochastic posterior is much broader than the Gaussian-like deterministic posterior. It is clear that for the stochastic posterior, the integration must extend considerably further into the tail,

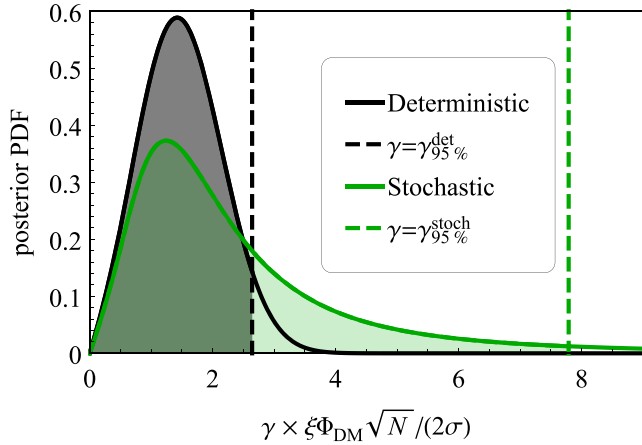

**Fig. 2 Posterior distributions for the coupling strength $\gamma$ in the deterministic and stochastic treatments, Eqs. (5) and (6), respectively.** Due to the fat-tailed shape of the stochastic posterior one can clearly see the 95% limit is larger with $\gamma_{95\%}^{stoch}/\gamma_{95\%}^{det} \approx 3.0$. The assumed value of the data is at the 95% detection threshold $A^{dt} = \sqrt{-\ln(1 - 0.95)}$ (see text).

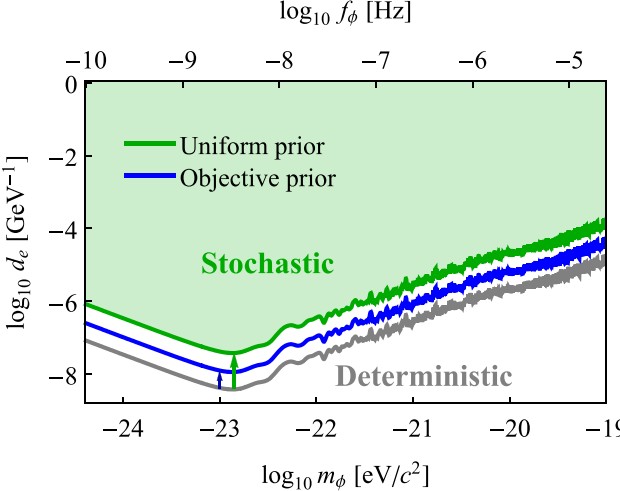

**Fig. 3 The modified constraint, green and blue lines, based on the stochastic approach compared to previous laboratory constraints, gray line, based on the deterministic approach for the dilaton coupling strength $d_e$[41, 46-48].** The green and blue lines illustrate the importance of the choice of prior for a Bayesian approach. Supplementary Fig. 3 provides a detailed exclusion plot.

leading to larger values of $\gamma_{95\%}$ and thereby to weaker constraints, $\gamma_{95\%}^{\text{stoch}} > \gamma_{95\%}^{\text{det}}$. Explicit evaluation of Eq. (4) with the derived posteriors results in a relation between the constraints

$$\gamma_{95\%}^{\text{stoch}} \approx 3.0 \ \gamma_{95\%}^{\text{et}} \ , \tag{7}$$

where the numerical value of the correction factor depends on CL and assumed value of $A$ (the factor increases for higher CL and decreases for smaller $A$).

This correction factor becomes $\approx 10$ when derived using a uniform prior, as shown in Supplementary Note 1. However, the result obtained with the uniform prior is not invariant under a change of variables (e.g., from excess amplitude to power). Additionally, using the objective prior yields better agreement with frequentist-based results of a factor $\approx 2.7$. For the gradient coupling of pseudoscalar particles, the directional sensitivity, deterministic assumptions, and initial phase of the field (when relevant) can further impact this factor as discussed in Supplementary Note 1.

## Discussion

Ultralight DM candidates are theoretically well-motivated and an increasing number of experiments are searching for them. Most of the experiments with published constraints thus far are haloscopes, sensitive to the local galactic DM and affected by Eq. (7). However, experiments that measure axions generated from a source, helioscopes, or new-force searches, for example, do not fall under the assumptions made here. We illustrate how the existing constraints have been affected in Fig. 3 and provide a more detailed exclusion plots for dilaton couplings[46-48] in Supplementary Note 2.

To interpret the results of an experiment searching for bosonic DM in the regime of measurement times smaller than the coherence time, stochastic properties of the net field must be taken into account. An accurate description accounts for the Rayleigh-distributed amplitude $\Phi_0$, where the variation is induced by the random phases of individual virialized fields. Accounting for this stochastic nature yields a correction factor of $\approx 2.7-10$, relaxing existing experimental bosonic DM constraints in this regime. In the event of a bosonic DM discovery, the stochastic properties of the field would result in increased uncertainty in the

determination of coupling strength or local average energy density in this regime.

It is important to note that observational knowledge of the energy distribution of DM[81] could constrain the stochastic behavior of the amplitude (or energy density). The smallest features observed so far are on the order of $\approx 0.1$ kpc[82] (corresponding to a coherence length of a boson of mass $m_\phi \approx 10^{-22}$ eV), however the analysis in ref.[82] performs angular averages which would suppress the stochastic variation discussed in this paper.

## Data availability

All conclusions made in this paper can be reproduced using the information presented in the manuscript and/or Supplementary Information. Additional information is available upon reasonable request to the corresponding author. For access to the experimental data presented here please contact the corresponding authors of the respective papers.

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

## Acknowledgements

We thank Eric Adelberger and William A. Terrano for pointing out the need to account for the unknown phase in the CASPEr-ZULF Comagnetometer analysis. We thank Kent Irwin, David J. E. Marsh, Lam Hui, Marina Gil Sendra, and Martin Engler for helpful discussions and suggestions. We thank M. Zawada, N. A. Leefer, and A. Hees for providing raw data for the published deterministic constraints. We also thank Jelle Aalbers for helpful discussions and expert advice on the blueice inference framework. Jan Conrad appreciates the support by the Knut and Alice Wallenberg Foundation. This project has received funding from the European Research Council (ERC) under the European Unions Horizon 2020 research and innovation program (Grant Agreement No. 695405). We acknowledge the partial support of the U.S. National Science Foundation, the Simons and Heising-Simons Foundations, and the DFG Reinhart Koselleck project.

## Author contributions

All authors have contributed to the publication. G.P.C., A.G., A.V.G., D.F.J.K., and A.D. contributed to derivations of dark matter properties and/or statistical results. G.P.C., J.W.B., J.C., N.L.F., M.L., B.P., J.A.S., A.O.S., A.W., and D.B. interpreted and processed data, constructed and implemented simulations, and contributed to key conceptual discussions. The manuscript was drafted by G.P.C., D.F.J.K., D.B., and A.D. All authors reviewed and approved the final version of the manuscript.

## Funding

## Competing interests

The authors declare no competing interests.

## Additional information

**Peer review information** *Nature Communications* thanks David Marsh and the other anonymous reviewer(s) for their contribution to the peer review this work. Peer reviewer reports are available.

