## [Peer Review File · Nature Communications]

Editorial Note:

This manuscript has been previously reviewed at another journal that is not operating a transparent peer review scheme. This document only contains reviewer comments and rebuttal letters for versions considered at *Nature Communications*.

Reviewers' Comments:

Reviewer #1:

Remarks to the Author:

I thank the authors for the new manuscript, and recommend its publication in *Nature Communications* with only some minor changes to the introductory sentences (see below, and annotated manuscript). I do not need to see these changes.

The authors did substantial new work to address the other reviewer's comments on the gradient coupling.

I thank the authors for the supplementary analysis including heavy tails and improved discussion around simulations.

I apologise for the confusion I created around comment 1, and reviewer 3. The new Gaia reference is excellent.

=====

Suggested changes:

I would change the new sentence "We note that most cosmological simulations include these stochastic properties [38,39]". I suggest:

"Cosmological simulations of VULF can and do resolve these stochastic properties~\cite{},"

The choice of phrasing is relevant. The simulations by Chan et al were not cosmological, but rather toy models of individual halos. On the other hand, *Most* cosmological simulations are N-body, and do not contain these effects. Indeed for something like the QCD axion, it would be impossible to resolve the effects in a cosmological simulation of the Milky Way. The qualifier "VULF" is necessary to say what type of simulations are referred to, and I think some qualifier is necessary around "resolve", since this fixes the boson mass and system size where the features can indeed be seen. If you want to keep the qualifier "cosmological" prior to "simulations" I would suggest from the current list only [39,40].

More generally "simulations" could include [38,46] also, but they are toy models. One could add further qualifiers like "fully non-linear and gravitating simulations" or some such, or "astrophysical and cosmological" if you want to widen the net.

David J. E. Marsh

Reviewer #3:

Remarks to the Author:

The paper ``Stochastic fluctuations of bosonic dark matter" points out that searches for bosonic dark matter (DM) in laboratory experiments need to carefully consider the stochastic nature of the DM on length scales of order the axion coherence length. I previously reviewed this paper as ``external reviewer \# 3" for *Nature Physics*.

I am happy with all of the changes made by the referees to their manuscript. I do, however, feel that the ``uniform prior" curve in Fig. 3 should be removed, as I find it to be misleading. I like how the authors now show that the objective prior leads to very similar results to the frequentist

results. Frequentist methods are more common in this field because it is hard to assign a prior expectation to unknown couplings. The objective prior has the nice attribute, as the authors explain, of being somewhat agnostic to this prior expectation, and it is very reassuring that it matches the more standard frequentist result. (As a side-note: I would recommend presenting the frequentist results in the main body and not the Bayesian results for these reasons, though I leave this decision to the authors.) The linear prior, on the other hand, is I think very unjustified, as it biases the coupling to large coupling strengths, which in turn is responsible for the large difference seen in Fig. 3. I think it is fine for the authors to have an appropriately caveated discussion of the linear prior in the text, but to avoid confusion for those skimming the paper I would strongly recommend removing this from Fig. 3. Including the uniform prior result also makes it hard to understand the point of Fig. 3, as looking at that figure it appears that the prior choice is more important than the correct description of the stochastic background (which is not the point I feel that the authors are trying to make).

As a minor point --- I'm not sure if the sentence beginning with ``Note that a preprint of this paper .." is appropriate (I would recommend removing this, to avoid closed literature loops), but I would leave this up to the editor.

At this point, I am happy with this paper, and I believe that it deserves publication in its current form somewhere. I do not feel that Nature Communications is the correct journal, however, though I leave this decision up to the editor. My reasoning is similar to that which I described in my previous response for Nature Physics (similarly, Nature Communications is a broad journal that reaches a broad scientific cross-section, and thus results published in this journal should be truly transformational). I do not think these results are largely transformational (though important in detail for specific experiments) for a number of reasons. Primarily, I am concerned that this method only applies to experiments that are probing tuned regions of beyond the standard model parameter space, even if such experiments exist and will make use of this method. With that in mind, a factor ~ 3 change in the upper limits is not a ground-breaking discovery (again, I agree it is important to get this correct in detail), as it would be for e.g. the QCD axion (if this paper was e.g. claiming a factor of 3 change in the QCD axion limit, I would certainly recommend publication in Nature Communications, but it is not), since the parameter space being searched over by the experiments under discussion has no theoretically motivated target coupling strength.

In their response the authors point to the recent paper 1907.03767 as an example of a laboratory experiment for which this paper plays an important role that has sensitivity close to the astrophysical bounds on ALPs. The authors' claim, however, is very misleading. The astrophysical bounds make no assumption about the ALPs or light scalars being dark matter, while the laboratory constraints require the axion (or scalar) to be dark matter. What is deceptive is that it is incredibly difficult to write down a model where at these masses and coupling strengths the axion (or scalar) is a non-trivial fraction of the dark matter (perhaps the authors know of a model?). The misalignment mechanism, for example, would predict virtually no dark matter fraction at these masses. Thus, it is in some sense very deceptive to even put the laboratory constraints (which require dark matter) and the astrophysical constraints (which do not) on the same plot. This mix of astrophysical and laboratory constraints is more common when looking at the higher-mass, QCD axion part of the parameter space since here it is more natural to expect the axion to be dark matter. The same complaint applies to the dilaton models. The authors mention that fuzzy dark matter is well motivated around 10^{-22} eV, and while this is true it is important to remember that the fuzzy dark matter couplings to matter are GUT-scale suppressed, and thus much smaller than what any of the laboratory experiments is claiming they can do in the future.

As the authors say, there exist laboratory experiments who should read this paper to correctly display their limits. But these experiments (those for which this paper applies to) do not probe motivated parameter space in that there do not exist models (at least, none that I am aware of) that may naturally produce the dark matter abundance in this coupling and mass range: these masses and couplings, as far as I am aware, vastly underpredict the observed dark matter. Therefore, I do not see this work as being relevant to the broader physics community and especially not to the broader scientific community. This is why I recommend this paper for a more specialized journal.

Response to third round of editorial assessment and reviewers' comments

November 10, 2021

Dear Editors of Nature Communications,

We are extremely happy to hear of our manuscript's acceptance to Nature Communications after this third round of reviewers. Our response and changes are outlined point-by-point below. We are happy to make any further changes the editors deem necessary.

Notes on response formatting

The entire review from each referee is provided in blue, with our response in black. Comments that dictated or suggested changes to the manuscript are reemphasized and numbered:

(#) Reviewer comment example.

Changes made to the paper that can be easily displayed here are denoted using:

Example excerpt from paper material that now addresses a comment.

Response to external reviewer #1:

I thank the authors for the new manuscript, and recommend its publication in Nature Communications with only some minor changes to the introductory sentences (see below, and annotated manuscript). I do not need to see these changes.

The authors did substantial new work to address the other reviewer's comments on the gradient coupling. I thank the authors for the supplementary analysis including heavy tails and improved discussion around simulations. I apologise for the confusion I created around comment 1, and reviewer 3. The new Gaia reference is excellent.

Suggested changes:

I would change the new sentence "We note that most cosmological simulations include these stochastic properties [38,39]". I suggest:

"Cosmological simulations of VULF can and do resolve these stochastic properties [],"

The choice of phrasing is relevant. The simulations by Chan et al were not cosmological, but rather toy models of individual halos. On the other hand, *Most* cosmological simulations are N-body, and do not contain these effects. Indeed for something like the QCD axion, it would be impossible to resolve the effects in a cosmological simulation of the Milky Way. The qualifier "VULF" is necessary to say what type of simulations are referred to, and I think some qualifier is necessary around "resolve", since this fixes the boson mass and system size where the features can indeed be seen. If you want to keep the qualifier "cosmological" prior to "simulations" I would suggest from the current list only [39,40].

More generally "simulations" could include [38,46] also, but they are toy models. One could add further qualifiers like "fully non-linear and gravitating simulations" or some such, or "astrophysical and cosmological" if you want to widen the net.

We would like to again thank David J. E. Marsh for his excellent insight throughout the review process and appreciate his help with improving our manuscript. We have implemented all the recommended changes, as outlined below, and hope he is content with the final manuscript to be published in Nature Communications.

Reviewer #1, Comment 1. I would change the new sentence "We note that most cosmological simulations include these stochastic properties [38,39]".

We revise the sentence in the manuscript as suggested to "widen the net" and maintain the current references.

We note that some astrophysical and cosmological simulations can and do resolve these stochastic properties [4, 17], however in this paper we discuss their impact on inferences drawn from direct detection experiments.

In addition to the reviewer response above there were a few changes to references recommended on the annotated manuscript we received. To summarize the included changes:

1. We added the reference regarding Bose Einstein condensate formation [11] as suggested.
2. Added the clause regarding axions stars "... small-scale structure such as miniclusters and axion stars [7, 16, 10]. Where we fixed the references as suggested.
3. We expanded the list of references regarding "the statistical properties of the VULF" to include Refs. [12, 16, 9, 15].

Response to external reviewer #3:

The paper “Stochastic fluctuations of bosonic dark matter” points out that searches for bosonic dark matter (DM) in laboratory experiments need to carefully consider the stochastic nature of the DM on length scales of order the axion coherence length. I previously reviewed this paper as “external reviewer # 3” for Nature Physics.

I am happy with all of the changes made by the referees to their manuscript. I do, however, feel that the “uniform prior” curve in Fig. 3 should be removed, as I find it to be misleading. I like how the authors now show that the objective prior leads to very similar results to the frequentist results. Frequentist methods are more common in this field because it is hard to assign a prior expectation to unknown couplings. The objective prior has the nice attribute, as the authors explain, of being somewhat agnostic to this prior expectation, and it is very reassuring that it matches the more standard frequentist result. (As a side-note: I would recommend presenting the frequentist results in the main body and not the Bayesian results for these reasons, though I leave this decision to the authors.) The linear prior, on the other hand, is I think very unjustified, as it biases the coupling to large coupling strengths, which in turn is responsible for the large difference seen in Fig. 3. I think it is fine for the authors to have an appropriately caveated discussion of the linear prior in the text, but to avoid confusion for those skimming the paper I would strongly recommend removing this from Fig. 3. Including the uniform prior result also makes it hard to understand the point of Fig. 3, as looking at that figure it appears that the prior choice is more important than the correct description of the stochastic background (which is not the point I feel that the authors are trying to make).

As a minor point — I’m not sure if the sentence beginning with “Note that a preprint of this paper ..” is appropriate (I would recommend removing this, to avoid closed literature loops), but I would leave this up to the editor.

At this point, I am happy with this paper, and I believe that it deserves publication in its current form somewhere. I do not feel that Nature Communications is the correct journal, however, though I leave this decision up to the editor. My reasoning is similar to that which I described in my previous response for Nature Physics (similarly, Nature Communications is a broad journal that reaches a broad scientific cross-section, and thus results published in this journal should be truly transformational). I do not think these results are largely transformational (though important in detail for specific experiments) for a number of reasons. Primarily, I am concerned that this method only applies to experiments that are probing tuned regions of beyond the standard model parameter space, even if such experiments exist and will make use of this method. With that in mind, a factor ~ 3 change in the upper limits is not a ground-breaking discovery (again, I agree it is important to get this correct in detail), as it would be for e.g. the QCD axion (if this paper was e.g. claiming a factor of 3 change in the QCD axion limit, I would certainly recommend publication in Nature Communications, but it is not), since the parameter space

being searched over by the experiments under discussion has no theoretically motivated target coupling strength.

In their response the authors point to the recent paper 1907.03767 as an example of a laboratory experiment for which this paper plays an important role that has sensitivity close to the astrophysical bounds on ALPs. The authors' claim, however, is very misleading. The astrophysical bounds make no assumption about the ALPs or light scalars being dark matter, while the laboratory constraints require the axion (or scalar) to be dark matter. What is deceptive is that it is incredibly difficult to write down a model where at these masses and coupling strengths the axion (or scalar) is a non-trivial fraction of the dark matter (perhaps the authors know of a model?). The misalignment mechanism, for example, would predict virtually no dark matter fraction at these masses. Thus, it is in some sense very deceptive to even put the laboratory constraints (which require dark matter) and the astrophysical constraints (which do not) on the same plot. This mix of astrophysical and laboratory constraints is more common when looking at the higher-mass, QCD axion part of the parameter space since here it is more natural to expect the axion to be dark matter. The same complaint applies to the dilaton models. The authors mention that fuzzy dark matter is well motivated around 10^{-22} eV, and while this is true it is important to remember that the fuzzy dark matter couplings to matter are GUT-scale suppressed, and thus much smaller than what any of the laboratory experiments is claiming they can do in the future.

As the authors say, their exist laboratory experiments who should read this paper to correctly display their limits. But these experiments (those for which this paper applies to) do not probe motivated parameter space in that there do not exist models (at least, none that I am aware of) that may naturally produce the dark matter abundance in this coupling and mass range: these masses and couplings, as far as I am aware, vastly underpredict the observed dark matter. Therefore, I do not see this work as being relevant to the broader physics community and especially not to the broader scientific community. This is why I recommend this paper for a more specialized journal.

We thank the reviewer for taking another look at our manuscript and are glad to hear they approve of the changes made. We address their further concerns below.

Reviewer #3, Comment 1 I like how the authors now show that the objective prior leads to very similar results to the frequentist results. [...] The linear prior, on the other hand, is I think very unjustified, as it biases the coupling to large coupling strengths, which in turn is responsible for the large difference seen in Fig. 3.

The reviewer's second paragraph mainly deals with our inclusion of Bayesian inference and discussion of the prior distribution assumed for the parameter of interest, being the coupling strength. We largely agree with the reviewer's point of view. However, we feel that this inclusion is quite important because it is exactly this class of problem (as outlined in Berger-Bernardo [2] Sec. 4) that

leads to the large discrepancies in the derived constraints when compared to Frequentist methods (the scalar correction term becomes 10 or greater and the gradient correction can easily exceed an order of magnitude). So this rather pedantic detail of Bayesian inference remains quite important when comparing constraints - coincidentally about as important as including the stochastic nature of the dark matter field in the first place.

In a way the reviewer further motivates our inclusion of this Bayesian detail with their concern regarding the linear prior, “which biases the coupling to larger strengths.” We would like to point out that Jefferys’ prior defines a distribution that minimizes the impact of the choice of variables on the derived constraint. While the uniform prior may seem natural to choose (naively assuming equal probability for each coupling strength), as we discuss in the paper, this uniform probability impacts the constraint in a nonphysical way. For example, if one chooses to work with their data in amplitude spectral density versus power. So assuming a uniform prior (also known as an uninformed prior) for both an ASD and PSD analysis effectively weights coupling strengths differently for each case (exactly what one naively tries to avoid when assuming the uniform prior in the first place).

Reviewer #3, Comment 2 I do, however, feel that the “uniform prior” curve in Fig. 3 should be removed, as I find it to be misleading [...] I think it is fine for the authors to have an appropriately caveated discussion of the linear prior in the text, but to avoid confusion for those skimming the paper I would strongly recommend removing this from Fig. 3. Including the uniform prior result also makes it hard to understand the point of Fig. 3, as looking at that figure it appears that the prior choice is more important than the correct description of the stochastic background (which is not the point I feel that the authors are trying to make).

Regarding the comment of including the choice of prior in Fig. 3, we find that the Bayesian and Frequentist results agree (with either a uniform or informed prior) until the inclusion of a distributed nuisance parameter (e.g. the amplitude as discussed in the paper). Given the magnitude of impact this choice of prior has on the derived constraint we believe, while not as exciting, it is similarly important to the stochastic nature of the field itself to ensure constraints are comparable in future works. We prefer to leave this curve in and present our results as is, given our reasons outlined in our response to Comment 1 and 2.

To avoid any confusion, as the reviewer points out, we expanded on the Fig. 3 caption:

The green and blue lines illustrate the importance of the choice of prior for a Bayesian approach.

Additionally, we expand the discussion in the main text to further emphasize the issues with the uniform prior:

Results from Bayesian inference are sensitive to the choice of prior [1], and we find better agreement with frequentist based approaches when using an objective prior rather than a uniform prior $p(\gamma) = 1$ (as shown in Supplementary Note 1.A.). Additionally, the uniform prior yields constraints which are noninvariant under a change of variables.

Reviewer #3, Comment 3 Primarily, I am concerned that this method only applies to experiments that are probing tuned regions of beyond the standard model parameter space, even if such experiments exist and will make use of this method. With that in mind, a factor ~ 3 change in the upper limits is not a ground-breaking discovery (again, I agree it is important to get this correct in detail), as it would be for e.g. the QCD axion (if this paper was e.g. claiming a factor of 3 change in the QCD axion limit, I would certainly recommend publication in Nature Communications, but it is not), since the parameter space being searched over by the experiments under discussion has no theoretically motivated target coupling strength.

The remainder of the reviewers response is largely concerned with the motivation for the parameter space our particular regime deals with (bosonic dark matter with $T \lesssim \tau_c$, starting around $\lesssim 10^{-13}$ eV). In contrast, some of our theorist colleagues consider this region of parameter space quite motivated. Here are a few publications that show models in this mass range [9, 5, 3]. We would like to reiterate our previous response that whether or not the particular exclusions cited in this paper reject a known model is, in our humble opinion, beside the point. In any case, these limits are the best laboratory results in the mass range, and perhaps overall depending on the assumptions made for the astrophysical limits (which are a rough estimate [6]). We hope that this paper informs future experimentalists (including the next iterations of the experiments presented) from incorrectly excluding a known model at a weaker coupling.

Reviewer #3, Comment 4 In their response the authors point to the recent paper 1907.03767 as an example of a laboratory experiment for which this paper plays an important role that has sensitivity close to the astrophysical bounds on ALPs. The astrophysical bounds make no assumption about the ALPs or light scalars being dark matter, while the laboratory constraints require the axion (or scalar) to be dark matter. What is deceptive is that is incredibly difficult to write down a model where at these masses and coupling strengths the axion (or scalar) is a non-trivial fraction of the dark matter (perhaps the authors know of a model?).

We do indeed make the assumption that all the dark matter is made up of this bosonic candidate. The constraints in the field to date do this as well. As we point out in our response to the previous comment there are indeed models in this mass range.

We again thank the reviewer for helping to improve our paper and hope our decision to maintain the discussion of priors in the main text and in the figure is clear. We hope the changes made help avoid any confusion and have expanded the references regarding ultra-light dark matter candidates [13, 14, 8, 9, 5, 3] to emphasize the motivation of this mass range.

We are happy to make any suggested changes if the editors find it necessary.

Sincerely,

Gary P. Centers and coauthors

References

- [1] J. O. Berger and J. M. Bernardo. Ordered Group Reference Priors with Application to the Multinomial Problem. *Biometrika*, 1992.
- [2] J. M. Bernardo. Reference Posterior Distributions for Bayesian Inference. *Journal of the Royal Statistical Society: Series B (Methodological)*, 41(2):113–128, jan 1979.
- [3] D. Brzemiński, Z. Chacko, A. Dev, and A. Hook. Time-varying fine structure constant from naturally ultralight dark matter. *Physical Review D*, 104(7):075019, oct 2021.
- [4] J. H. H. Chan, H.-Y. Schive, T.-P. Woo, and T. Chiueh. How do stars affect ψ DM haloes? *Monthly Notices of the Royal Astronomical Society*, 478(2):2686–2699, aug 2018.
- [5] J. A. Dror, K. Harigaya, and V. Narayan. Parametric resonance production of ultralight vector dark matter. *Physical Review D*, 99(3):035036, feb 2019.
- [6] P. W. Graham and S. Rajendran. New observables for direct detection of axion dark matter. *Physical Review D - Particles, Fields, Gravitation and Cosmology*, 88(3):1–13, 2013.
- [7] C. Hogan and M. Rees. Axion miniclusters. *Physics Letters B*, 205(2-3):228–230, apr 1988.
- [8] W. Hu, R. Barkana, and A. Gruzinov. Fuzzy Cold Dark Matter: The Wave Properties of Ultralight Particles. *Physical Review Letters*, 85(6):1158–1161, aug 2000.
- [9] L. Hui, J. P. Ostriker, S. Tremaine, and E. Witten. Ultralight scalars as cosmological dark matter. *Physical Review D*, 2017.
- [10] D. F. Jackson Kimball, D. Budker, J. Eby, M. Pospelov, S. Pustelny, T. Scholtes, Y. V. Stadnik, A. Weis, and A. Wickenbrock. Searching for axion stars and Q -balls with a terrestrial magnetometer network. *Physical Review D*, 2018.
- [11] D. G. Levkov, A. G. Panin, and I. I. Tkachev. Gravitational Bose-Einstein Condensation in the Kinetic Regime. *Physical Review Letters*, 121(15):151301, oct 2018.
- [12] S.-C. Lin, H.-Y. Schive, S.-K. Wong, and T. Chiueh. Self-consistent construction of virialized wave dark matter halos. *Physical Review D*, 97(10):103523, may 2018.
- [13] D. J. Marsh. Axion cosmology. *Physics Reports*, 643:1–79, jul 2016.
- [14] D. J. Marsh and J. Silk. A model for halo formation with axion mixed dark matter. *Monthly Notices of the Royal Astronomical Society*, 2014.

- [15] D. J. E. Marsh and J. C. Niemeyer. Strong Constraints on Fuzzy Dark Matter from Ultrafaint Dwarf Galaxy Eridanus II. *Physical Review Letters*, 123(5):051103, jul 2019.
- [16] C. A. J. O’Hare and A. M. Green. Axion astronomy with microwave cavity experiments. *Physical Review D*, 95(6):063017, mar 2017.
- [17] J. Veltmaat, J. C. Niemeyer, and B. Schwabe. Formation and structure of ultralight bosonic dark matter halos. *Physical Review D*, 98(4):043509, aug 2018.